# Bio-Design, Fabrication and Analysis of a Flexible Valve

**DOI:** 10.3390/biomimetics7030095

**Published:** 2022-07-14

**Authors:** Zirui Liu, Bo Sun, Jiawei Xiong, Jianjun Hu, Yunhong Liang

**Affiliations:** 1State Key Laboratory of Mechanical Transmissions, Chongqing University, Chongqing 400044, China; 20162252@cqu.edu.cn; 2School of Mechanical and Aerospace Engineering, Jilin University, Changchun 130022, China; sunb19@mails.jlu.edu.cn; 3Key Laboratory of Bionic Engineering, Ministry of Education, Jilin University, Changchun 130022, China; liangyunhong@jlu.edu.cn

**Keywords:** biofluid control and actuation, bionic design, flexible fluid control valve

## Abstract

Fluid-driven soft robots offer many advantages over robots driven by other means in terms of universal preparation processes and high-power density ratios, but are largely limited by their inherit characteristics of rigid pressure sources, fluid control elements and complex fluid pipelines. In this paper, inspired by the principle of biofluid control and actuation, we combine simulation analysis and experimental validation to conduct a bionic design study of an efficient flexible fluid control valve with different actuation diaphragm structures. Under critical flexural load, the flexible valve undergoes a continuous flexural instability overturning process, generating a wide range of displacements. The sensitivity of the flexible valve can be improved by adjusting the diaphragm geometry parameters. The results show that the diaphragm wall thickness is positively correlated with the overturning critical pressure, and the radius of curvature is negatively correlated with the overturning critical pressure. When the wall thickness of the flexible valve maintains the same value, as the radius of curvature increases, the critical buckling load and recovery load of diaphragm overturning is a quadratic function of opposite opening, and the pressure difference converges to the minimum value at the radius of curvature R = 7.

## 1. Introduction

Soft robots are an emerging field with the advantage of flexibility and environmental adaptability driven by fluids or smart materials. They can achieve variable stiffness [1,2], passive energy storage and large strain characteristics by changing the structure and composition of the flexible materials [3,4,5,6]. The fluid actuator (pneumatic or hydraulic) is one of the most important actuators for soft robots because of its significant flexible deformation, low manufacturing costs and mechanical programmability [7]. It can replace rigid structures to achieve various mechanical motions such as rotation, rolling and grasping [8]. However, the lack of responsiveness of the actuator due to the external energy supply, material and fluid viscosity limits its application in pumps, valves and other fast cycling conditions.

The key control element (valve) of a fluid actuator is still being explored. At present, there are three common approaches to improve the response speed of fluid actuators. First, the response pressure can be obtained by a chemical combustion reaction, so as to significantly improve the response speed, which requires high performance materials [9,10]. Secondly, some scholars have adopted external high voltage or strong magnetic field to cause transient phase changes in materials, which can reduce the response time to the level of 10^−3^ s, but this method is more costly [11,12]. Finally, some have achieved this by mobilizing the structural instability of elastic materials including wrinkling, folding [13,14] and kinking [15] of structures affected by pressure or vacuum instabilities [16]. Designs involving fluid elastic structure instabilities as switching valves are common in recent years, as the programmability of fluid elastic structures has been heavily studied [17,18]. Previous related work has focused on biological valves, which regulate flexible lines by using higher fluid pressures to achieve fluid pumping [19]. Only a few studies have been conducted on the combination of the above-mentioned switching valves with soft pumping fluid systems that provide the drive energy.

Inspired by the rapid energy storage and release design of soft robots [20,21,22], a bionic soft valve consisting of two membrane actuators arranged vertically on one side of an elastic hose with fast response characteristics has been prepared. The operating principle is that with the injection of compressed gas, the elastic fluid cavity expands, which drives the bistable diaphragm upwards. When the pressure in the cavity reaches a critical load, the diaphragm flexes and deforms to limit the diameter of the flexible hose inside the valve, thereby closing the valve and providing a fluid shut-off seal. Combined with the fast-turning characteristics of the bistable diaphragm, a cyclic change in the diameter of the flexible hose is achieved by supplying compressed gas to the cycle, resulting in a reciprocating valve from closed to open. Simulation analysis and experimental tests were carried out to select a drive diaphragm with good sensitivity and energy utilization efficiency. The results show that the bionic soft valve provided is not only simpler in terms of process fabrication, but also has improved sealing pressure-bearing properties as well as responsiveness, in comparison with the results of some similar experiments [22,23,24,25].

This innovation of bionic soft valves will largely facilitate the development of soft robots towards a wider range, and may also find broad applications in scenarios such as operating systems for remote surgery [26] and wearable systems involving breathing [27] by its efficient control properties. In the future, research on the assembly line learning and optimization combined with deep learning will be more helpful to improve the performance of products [28,29].

## 2. Materials and Methods

Thin-walled spherical shells or spherical crown shell structures used in traditional mechanical fields in aerospace reservoirs are prone to buckling deformation when subjected to fluid pressure and quickly release energy through buckling jumps to achieve a rapid response in displacement output. Based on the reasonable configuration of its geometric structure parameters and material elastic modulus properties, the critical buckling pressure and bistable response characteristics of the above unstable structure can be regulated.

Therefore, the actuation response characteristics of the above structure are similar to the accumulation and release mechanism of elastic potential energy in the process of biological actuation using muscles, and this feature will be suitable as the driving element in the design of flexible valves to achieve the improvement of fluid control efficiency in the elastic tube. In this paper, a new bionic flexible valve structure design is developed, as shown in Figure 1. It mainly consists of a flexible fluid chamber with a drive diaphragm, a flexible hose for media flow and a flexible encapsulated housing. The flexible valve has a set of resilient fluid cavities with a driving diaphragm as the driving member of the valve, thus replacing the original mechanical valve piston and other structures, and the side wall of the fluid cavity is set with an inflatable line connected to the external pressure source. Herein, the design uses a casting process to manufacture the wall with a fixed limit hole encapsulated shell, and the inner wall of the shell and the fluid cavity outside diameter for the transition fit; the shell wall sets the limit hole, and it is easy to determine the relative assembly position of the elastic hose and elastic fluid cavity and other components; in addition, convex structures with annular tenons are provided at the upper and lower ends of the housing, which integrate in parallel due to subsequent expansion of the resilient valve.

The principle of the bionic soft valve is that the elastic fluid cavity increases in volume expansion after compressed gas injection, driving the diaphragm to move upward. When the cavity pressure increases to the critical load of the elastic diaphragm structure, the flexural jump deformation occurs, significantly changing the cross-sectional area of the pipe to limit the diameter of the elastic hose in the valve to achieve valve closure; as the pressure continues to increase, the fluid cut-off sealing effect is enhanced. Combined with the characteristics of the bistable diaphragm fast flip, the cyclic supply of compressed gas enables cyclic changes in the diameter of the flexible hose, resulting in a reciprocating state of the valve from closed to open.

The specific preparation process for flexible valves is as follows:

Easy release 200 spray is applied to the inner wall of the mold to ensure the integrity of the sample when it is separated from the mold. The Dragon Skin 30 silicone B: A components are thoroughly mixed at room temperature in a ratio of 1:1 by a stirrer. Afterwards, to ensure homogeneity of the components of the molded sample, the mixture is placed in a vacuum drying device for 5–10 min until the bubbles in the silicone rubber disappeared completely. The mold is assembled and fixed with bolts and nylon ties on the lower surface of the mold according to the 3D design model. In order to rend the casting sample with good molding size and no structural defects such as air holes, the mixed solution should be poured slowly until it is level with the upper edge of the mold and then covered with the upper cap. The photosensitive resin mold is inverted downward, using gravity to ensure that the pressure-bearing cavity prefabricated parts and flexible valve shell parts are well formed, and cured for about 16 h at rest. A small amount of silicone rubber solution is configured in the same proportion, and the liquid is poured into the sealing mold. The bottom of the mold and the outer wall surface of the pressure-bearing cavity are in transition fit. The obtained sample parts of the prefabricated-pressure-bearing cavity are inserted into the uncured silicone solution from top to bottom, and then, the sealing mold and the sample parts are put together in a constant temperature heating oven at 40 °C to accelerate the curing cycle. The molded prototype is removed and a silicone adhesive (Sil-Poxy Silicone) is applied to the over-fit vulcanized silicone tube and the fluid cavity receiver. The curing was accelerated by local heating to ensure good airtightness and pressure-bearing in the cavity, a second fixation was made to the pipe connection with self-locking nylon rolling tape at the end of curing. Then the sample is inflated with a syringe and the diaphragm is turned over normally to verify the good air tightness of the sample. The comparison before and after the inflation drive is shown in Figure 1B.

## 3. Results

### 3.1. Simulation and Analysis

To study the influence of structural parameters on the response characteristics of the driving diaphragm, the elastomeric fluid cavity structure is divided into two types of three-dimensional models with different driving diaphragms (spherical diaphragm, spherical crown diaphragm and flat diaphragm) and a bottom pressure-bearing cavity. The simulation analysis model is simplified for the actual structure, and the cavity does not have an inflation line.

The deformation cloud diagram and load-displacement curve of the driven diaphragm of spherical membrane with thickness 2 mm and radius 6 mm are shown in Figure 2A. The stress change of the diaphragm with displacement is shown in Figure 2B. From the cloud diagram, it can be seen that the eccentricity phenomenon is not produced during the process of spherical membrane turning, nor is there any fold, and the stress concentration area of turning is at the spherical membrane connection. The slope of the load displacement curve is the structural stiffness of the spherical membrane subjected to deformation. Before the pressure in the cavity first reaches its peak, the slope of the curve decreases slowly with increasing vertex displacement. At this point, the wall thickness of the ball membrane and sidewall connection is relatively thin, hence the flexible hinge connection appears first. As the pressure in the lumen increases to the peak point of the curve, the slope rapidly bottoms out to zero. At this point, the curvature of the spherical membrane in the dashed box continues to decrease until the membrane approaches a horizontal state. At the apex, the stiffness rapidly decreases to a value close to zero, and local instability buckling occurs. At the apex of the spherical membrane, the spherical membrane changes from upward “convexity” to downward “depression”, the spherical membrane changes from external pressure to internal pressure, and the stiffness rapidly decreases from positive to negative value during the spherical shell turning process. As the stiffness decreases in the region from the apex to spread around, the area under external pressure gradually transforms into the area under internal pressure, and the pressure load in the cavity is abruptly reduced by the sudden change in volume until it reaches the critical valley point, i.e., the second curve inflection point. At this point, the diaphragm completely flips over, and the inner wall surface of the diaphragm is transformed from pressure-bearing flexure to expansion to produce deformation. Thereafter, the ball membrane is again in a new steady state form, the slope stiffness increases instantly, while the expansion deformation generated by the diaphragm with the pressure growth slows down significantly and the sensitivity decreases.

It can be seen that the overturning process of the spherical crown membrane and the hemispherical membrane is similar, which is consistent with the second type of extreme point instability characteristics (Figure 3). The spherical crown membrane decreases rapidly after reaching the peak pressure (17.5 kPa), which is less than that of the spherical membrane. Combining with the cloud diagram, it can be seen that the spherical crown membrane has lower geometric stiffness and lower inversion stability compared with the spherical membrane. The slope of the curve decreases in a smaller range, indicating that the radius of curvature limits the fast response characteristics of the displacement generated by the flexural jump. Combined with the driving load-displacement relationship in Figure 3, the flat membrane differs from the spherical and crown membranes in that the membrane flip is stable and the maximum displacement appears at the center point of the membrane; meanwhile, as the load pressure increases, the output displacement is positively correlated with the input pressure, but its deformation displacement produced by flip under the same pressure is the smallest, so the flat membrane consumes more energy during the application process. A preliminary comparison of the three structures through simulation shows that the spherical membrane and spherical crown membrane structures have the characteristics of achieving a large displacement output at low driving pressure, while the spherical membrane structure requires larger tipping pressure for application and has relatively better tipping stability.

The effect of the diaphragm thickness factor on the load displacement response of the spherical and crown membranes with fast response characteristics is investigated by replacing the 3D model of the spherical membrane in the assembly analysis model for simulation. To facilitate the preparation and to ensure that the thickness of the sidewall surface is greater than the thickness of the diaphragm, three sets of wall thicknesses (1.5 mm, 1.8 mm, and 2 mm) are selected for each of the spherical and crown membrane structures, and the change of the diaphragm apex displacement during the increase of the intracavity loading pressure (PCAV) from 0 kPa to 40 kPa is obtained (Figure 4). As shown in Figure 4, the maximum vertical displacement of the diaphragm deformation under the same air pressure decreases with increasing the diaphragm wall thickness, and the critical pressure that represents energy consumption and ease of starting is negatively correlated with the wall thickness.

To investigate the effect of radius of curvature on the deformation of the driving diaphragm, the load-displacement relationship curves of three types of radius of curvature diaphragms were studied by setting the wall thickness of the spherical diaphragm at 1.5 mm, 1.8 mm and 2 mm, respectively, and the fluid pressure in the range of 0~40 kPa. As shown in Figure 5 both the spherical and crown diaphragms have lower starting pressures (15–30 kPa), and the critical flexural pressure at the inflection point decreases as the radius of curvature increases, i.e., the spherical crown diaphragm requires lower driving energy; at the same time, the pressure drop due to diaphragm flip decreases, i.e., the bistability characteristics decrease. Before each structure drives the diaphragm flip to reach the second steady-state inflection point, the same pressure is applied, and the larger the radius of curvature, the higher the displacement generated by the flip; while after the inflection point, the driving diaphragm sensitivity and the maximum flip displacement are negatively correlated with the radius of curvature, so that the spherical membrane flip has a greater application range.

### 3.2. Experimental Testing and Analysis

To investigate the deformation characteristics of the prepared flexible valve driving diaphragm under isovolumetric quasi-static loading conditions, a quasi-static experimental platform is built. As shown in Figure 6, the load and displacement data of the driving process are collected by sensors, and the physical diagram of the modal change is intercepted by a high-speed camera combined with PREMIERE. The control pressure and diaphragm flip rate in the fluid cavity of three structures, namely, spherical membrane, spherical crown membrane and flat membrane, are shown in Figure 7, Figure 8 and Figure 9 with driving time.

The transient pressure curves of the spherical diaphragm type drive diaphragm during loading and unloading were obtained by the pressure sensor. As shown in Figure 7, the pressure variation of the spherical diaphragm is basically continuous in accordance with the quasi-static test conditions. The test process is divided into three stages: loading, stopping and unloading, the critical pressure value of the spherical diaphragm changes twice during loading and unloading, and the maximum pressure change is 3.1 kPa at 12.5 s. To investigate the cause of the pressure change, we specified the flip rate as the ratio of the rise height of the point collected by the diaphragm laser to the maximum driving output displacement of the diaphragm. Combined with the physical diagram of flip-flop and the diaphragm flip-flop rate curve, it can be seen that the driving flip-flop of the diaphragm is affected by the flexible material and the pressure inside the cavity shows a non-linear relationship. During the first ~12.5 s, as the intracavity pressure increases, the slope of the curve represents an increase in the overturning rate (parabola with upward opening). When the intracavity pressure reaches the critical overturning pressure of the diaphragm, the ball membrane rapidly changes from “concave” to “convex”, and the overturning rate also changes from “concave” to “convex” within 12.5~12.6 s. When the intracavity pressure reaches the critical flip pressure of the diaphragm, the spherical membrane rapidly changes from “concave” to “convex” during 12.5~12.6 s, and the flip rate also changes from 27.6% to 97.2%. This is due to the rapid increase in the volume of the cavity during the rapid flip of the diaphragm from the bottom to the top, and therefore the local pressure decreases. Thereafter, as the pressure continues to increase, the slope of the curve does not slow down to the overturning maximum. The above results show that the spherical diaphragm structure has two different critical pressure values, i.e., it has a bistable effect, and the diaphragm can achieve a sudden change in the flip rate at the critical pressure, indicating that the group of driving diaphragms has a fast response characteristic at the corresponding pressure.

The transient pressure curve of the spherical crown membrane and the overturning rate curve are shown in Figure 8. From the figure, we can see that the spherical crown membrane and the spherical membrane loading overturning process are similar, and the same overturning rate slope decreases with increasing pressure. Although there is no significant pressure mutation point on the pressure time-varying curve, combined with the flip rate curve, we can draw the conclusion that there is a mutation at 6.2 s in the spherical crown membrane, and the flip rate of the spherical crown membrane also mutates from 43.9% to 58.3%. The transient pressure curve is similar to that of the spherical crown membrane, and there is no critical load jump point during the flexure process.

During the dynamic process of the diaphragm turning rapidly from the bottom to the top, due to the compressibility of the gas, the volume in the cavity increases rapidly, resulting in a sudden decrease of the pressure in the cavity, and the corresponding pressure value is the corresponding critical load. The quasi-static test shows that the spherical membrane and the spherical crown membrane both have two different critical pressures, i.e., bistable flip characteristics, indicating that they have fast response characteristics under the corresponding pressure. The initial comparison with the flat membrane (Figure 9) indicates that the bistable characteristics of the three structures are gradually diminished with the increase of the diaphragm curvature driving the diaphragm actuator when only the curvature is different, which is in accordance with the simulation results.

The relationship between the actuation pressure of the flexible valve and the valve seal pressure-bearing was tested. In the test process, first of all, the flexible valve was composed of a set of elastic fluid cavities, commercial elastic latex tubes into the control gas P_control_ to close the valve, and then into the elastic latex tube compressed gas, slowly adjusting the regulator to increase the pressure in the hose until the observation of the slow emergence of bubbles in the water tank; the pressure sensor readings for the flexible valve critical opening pressure P_open_ were recorded. Afterwards, we raised the pressure inside the tube until the bubbles appeared steadily, repeatedly adjusting the valve so that the pressure inside the hose gradually decreased until the sink stopped producing gas, and recorded the corresponding pressure value for the critical closing pressure P_closed_. The experiments were repeated three times, and the average values were taken to establish the relationship between the maximum sealing pressure of the valve and the control pressure. The valve critical pressure test and data processing were carried out by flexible valves prepared from three sizes of ball and crown membranes with bistable characteristics. The test results prove that when the driving pressure is the same, the flexible valve is in the closed state, and its control line first increases and then decreases during the process; the flexible valve breakthrough opening pressure P_open_ and again closed pressure P_closed_ aredifferent. There will be two critical pressures, that is, the flexible valve and the ball membrane/ball crown membrane actuator with similar bistability characteristics, and the opening pressure of the flexible valve of the same structure is always higher than the valve closing pressure. As displayed in Figure 10, the driving pressure of the flexible valves of the three structures is positively correlated with the controlled line pressure. When the drive control pressure is the same, the critical pressure of the ball membrane type is greater than that of the ball crown type flexible valve, and the curvature of the larger ball crown type flexible valve critical pressure difference is greater. This is due to the fact that when the wall thickness is certain, the larger the curvature, the smaller the flip displacement produced by the diaphragm under the same driving pressure. Therefore, the ball-diaphragm valve has better sealing performance compared with the ball-crown valve. As the curvature increases, the critical bistable range of the ball-crown valve (ΔP = P_open_ − P_closed_) also increases. The results indicate a better sealing effect for the same actuation force and a lower actuation energy required to keep the flexible valve closed, which validates the rationality of the bistable flexible valve manufacturing design.

## 4. Discussion

In this paper, we propose to use the driving diaphragm flip of the pressure flexural jump to release energy to deform the elastic fluid line as the design principle of the flexible valve. According to the simulation analysis of the drive diaphragm designed by the flexible valve, the results show that the spherical membrane and the spherical crown membrane structure will undergo a flexural jump overturning process when the drive pressure reaches the critical flexural load, with transient displacement response characteristics. The critical pressure comparison shows that the diaphragm wall thickness is positively correlated with the overturning critical pressure. The radius of curvature is negatively correlated with the overturning critical pressure, i.e., the flexible valve sensitivity can be improved by adjusting the diaphragm geometry parameters. By testing the pressurized actuation process of three types of actuated diaphragms (ball membrane R = 6 mm, crown membrane R = 7 mm, and flat membrane R = ∞), and by comparing the overturning rate, it is found that the flat membrane actuation varies linearly with time, and the response speed of the crown membrane is better than that of the ball membrane. Meanwhile, the driving and unloading processes of both spherical and spherical crown membranes are similar, in that the actuator diaphragm will flex and deform to release energy to achieve displacement mutation when the critical pressure is reached. In addition, the load-displacement curves show that all three diaphragms have bistability characteristics, and the comparison shows that the spherical diaphragm has a larger load flip range, i.e., a larger flip distance at a lower pressure. The wall thickness is positively correlated with the critical pressure value. For the same wall thickness, as the radius of curvature increases from 6 to 7 mm, the critical buckling load and the recovery load of the diaphragm overturning are quadratic functions with opposite opening, and the pressure difference converges to the minimum value at the radius of curvature R = 7.

Although the resultant bionic soft valve possesses obviously enhanced performances such as fast response and simple preparation, its lifetime and usage are still the key points to be overcome in the subsequent research. After extensive testing, it was found that its average life span is more than 2000 times, and the breakage location mainly appears on the wall where the pressure-bearing chamber is connected to the diaphragm. The cycle life and reliability of flexible valves will be improved to a certain extent by reasonably enhancing the lateral modulus of elasticity and structural optimization. In the long run, it will make more sense to improve the controllability of the soft valves by means of deep learning.

## Figures and Tables

**Figure 1 biomimetics-07-00095-f001:**
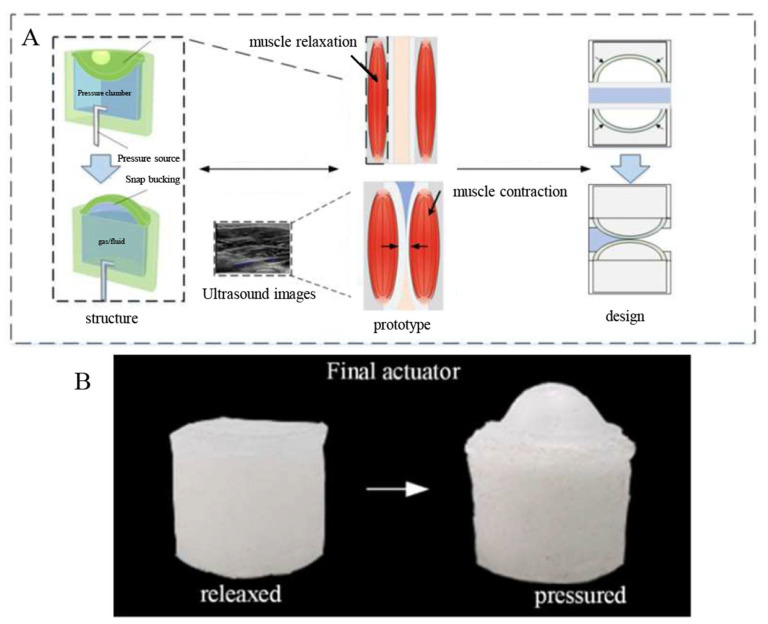
(**A**) Structure schematic of the new bionic flexible valve, (**B**) Physical drawing of the new bionic flexible valve.

**Figure 2 biomimetics-07-00095-f002:**
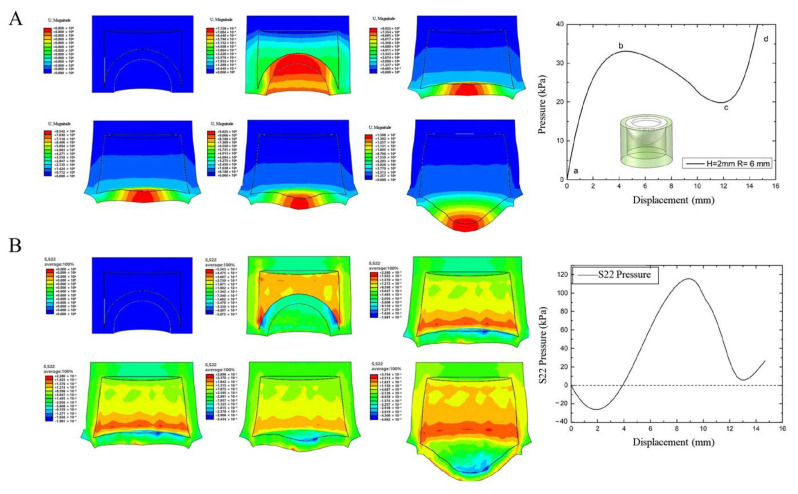
(**A**) The deformation cloud diagram and load-displacement curve of the driven dia-phragm, (**B**) The stress cloud diagram and the stress-displacement curve.

**Figure 3 biomimetics-07-00095-f003:**
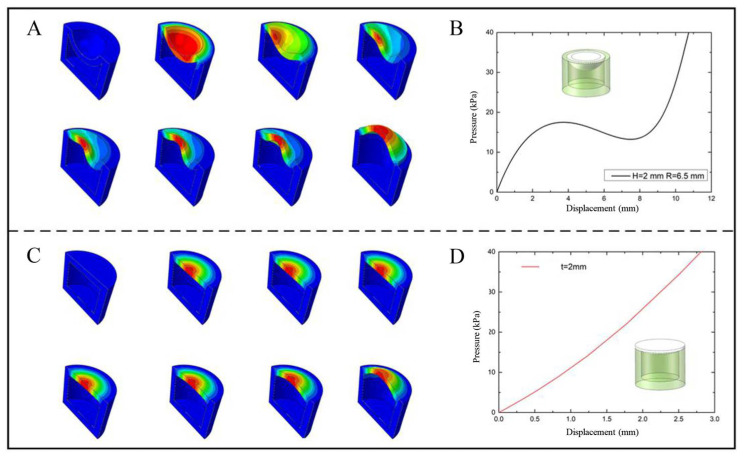
(**A**) The deformation cloud diagram of spherical crown membrane, (**B**) The load displacement curve of spherical crown membrane, (**C**) The deformation cloud diagram of flat membrane, (**D**) The load displacement curve of flat membrane.

**Figure 4 biomimetics-07-00095-f004:**
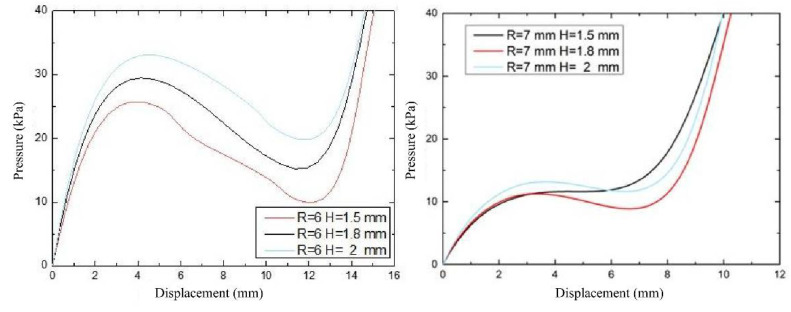
Relationship between spherical membrane and crown membrane load displacement under the influence of wall thickness factor.

**Figure 5 biomimetics-07-00095-f005:**
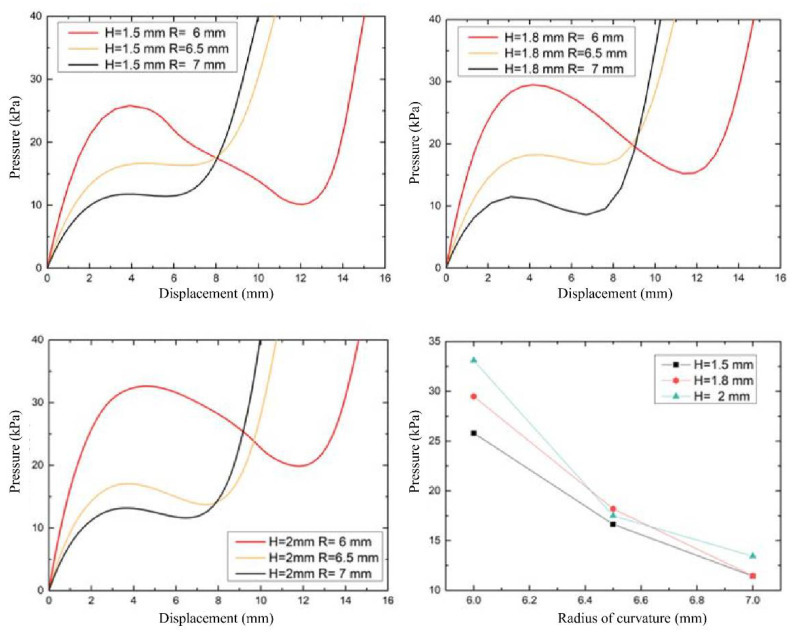
Load-displacement relationship under the influence of curvature factor.

**Figure 6 biomimetics-07-00095-f006:**
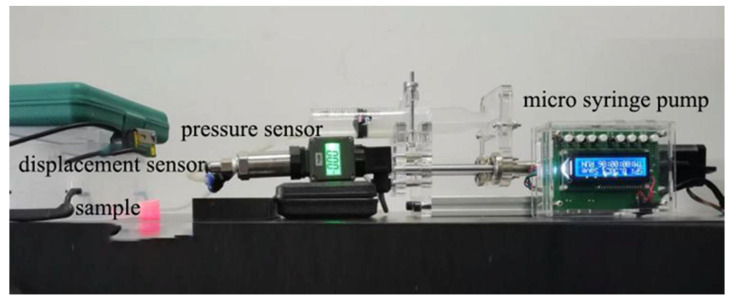
Physical diagram of diaphragm flip quasi-static test rig.

**Figure 7 biomimetics-07-00095-f007:**
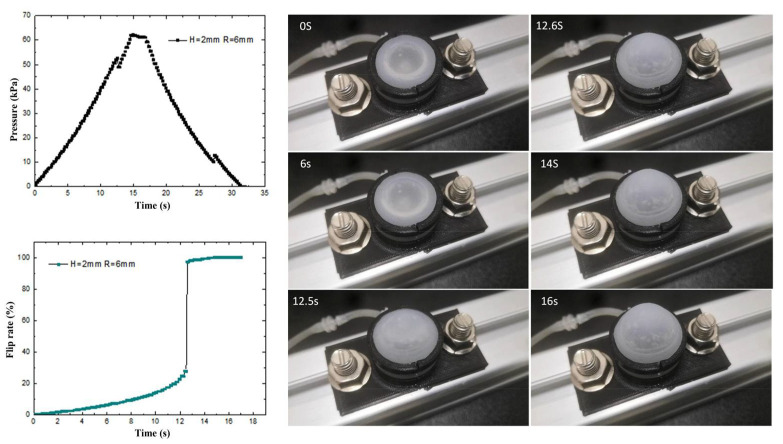
Ball membrane driving process, pressure time variation and loading flip rate curve.

**Figure 8 biomimetics-07-00095-f008:**
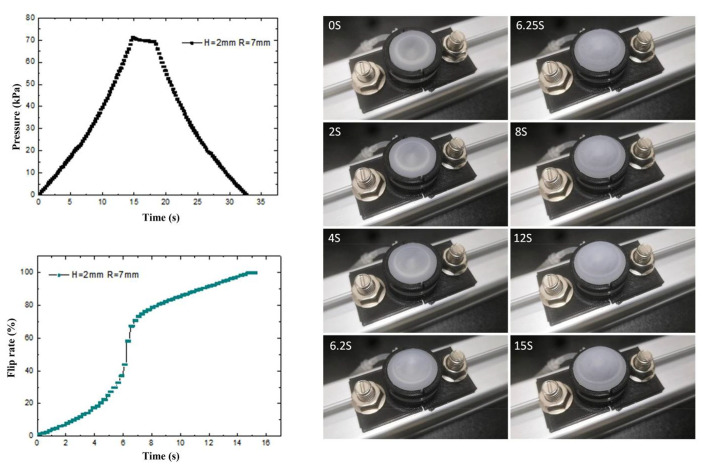
Ball crown membrane overturning process, pressure time variation and loaded overturning rate curve.

**Figure 9 biomimetics-07-00095-f009:**
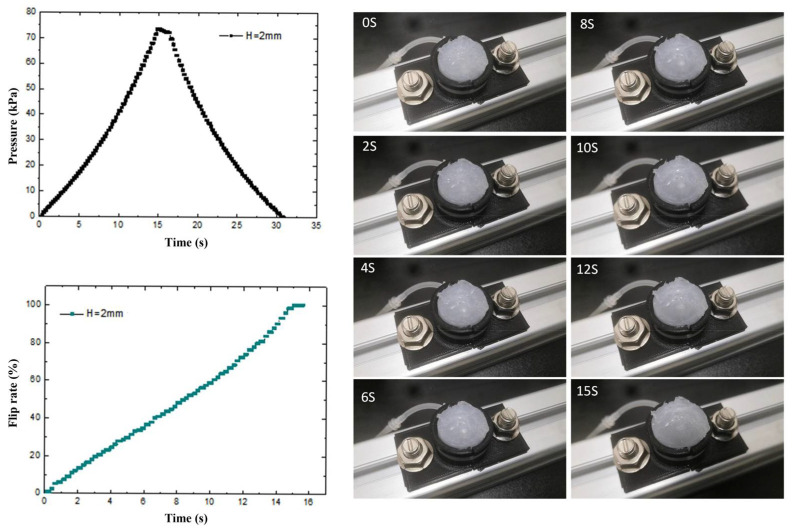
Curves of flat film driving process, pressure time variation and loading flip rate.

**Figure 10 biomimetics-07-00095-f010:**
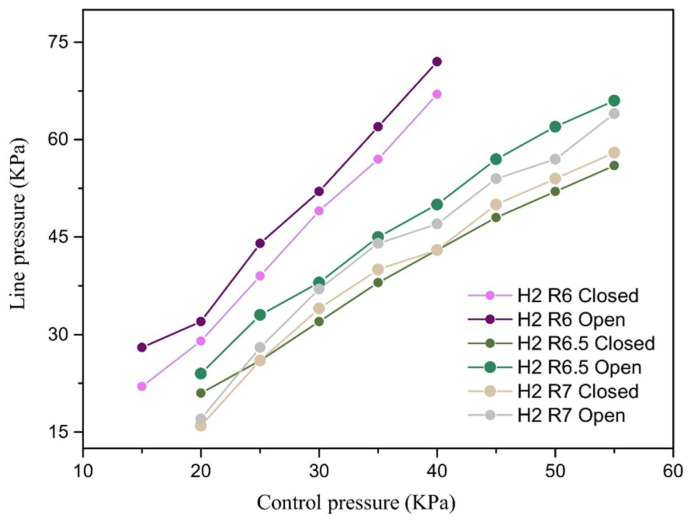
The relationship between driving pressure and line pressure for three types of flexible valves.

## Data Availability

The data presented in this study are available upon request from the corresponding author.

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
