# Peer review of "Bio-Design, Fabrication and Analysis of a Flexible Valve"

_biomimetics, 2022, doi:10.3390/biomimetics7030095_

Round 1

Reviewer 1 Report

In this manuscript, Liu et al. developed a flexible bistable valve for soft robotics applications. Its performance with different diaphragm structures were analyzed and its correlation with the diaphragm thickness was investigated. This paper has a clear structure and all results and figures are clearly illustrated. The reviewer has the following comments:

1.     The reviewer suggests the authors to improve the English writing of this paper. In its current form, this paper has a relatively poor readability and contains some grammar errors and typos. This paper also has some long paragraphs that affect the readability, please break into several shorter paragraphs.

2.     The novelty of this paper should be more clearly discusses in the introduction. In particular, the difference between this bistable valve design and the other soft values with the same principles and mechanisms (e.g. [20,22] in this paper), should be clearly explained.

3.     In Figure 10, these multiple lines with different colors are somewhat confusing. The reviewer recommends to use adjacent colors for the same design, e.g. H2 R6 Closed: Gray, H2 R6 Open: Black; H2 R6 Closed: Pink, H2 R6 Open: Red

Reviewer 2 Report

This paper focuses on the issue of soft robots driven limited by rigid pressure sources, fluid control elements, and complex fluid pipelines. The purpose of this paper is to combine simulation analysis and experimental verification to conduct a bionic design study of a new and efficient flexible fluid control valve with different actuation diaphragm structures. It is true that the flexible valve undergoes a continuous flexural instability overturning process under critical flexural loading, generating a wide range of displacements. The sensitivity of the flexible valve can be improved by adjusting the diaphragm geometry parameters. This research is interesting for the soft robot control research society. However, this paper has several limitations and the standard is not enough, and address the following items would result in a good paper,

1.  The literature review is not thorough about the application and the contributions. To highlight the contributions, it suggests reorganizing the section of the related work with real applications. It is recommended to read more related works and consider discussing their application scenarios in the introduction and discussion, such as, Fuzzy Approximation-based Task-Space Control of Robot Manipulators With Remote Center of Motion Constraint, A Multimodal Wearable System for Continuous and Real-time Breathing Pattern Monitoring During Daily Activity, etc.

2.  The contribution of this paper is not clear. It suggests revising the contributions section and making these points clear and strong.

3.  The quality of the Figures should be improved and readable for the readers.

4.  Maybe it is better to discuss the possibility to improve the scope using deep learning in assembly line to learn and optimize for online estimation in the introduction, for example, Multi-sensor guided hand gesture recognition for a teleoperated robot using a recurrent neural network; A Cybertwin based Multimodal Network for ECG Patterns Monitoring using Deep Learning,.

5.  It is recommended to present in the first section so that it can highlight the specific scope of this article. The meaning of the assessment experiment should be highlighted.

6.  Overall, proofreading is preferred. The current version is not written in a good and clear way. The English description should be improved and the grammar should be carefully addressed. It suggests consulting the publisher to polish your English writing, for example,   http://mugepaper.com or using https://www.ajdcets.com/ or other similar services.

7.  There should be a further discussion about the limitation of the current works, in particular, what could be the challenge for its related applications. To let readers better understand future work, please give specific research directions.

Round 2

Reviewer 2 Report

The authors have addressed all of my concerns. The current version can be accepted now.